# A New Phased-Array Magnetic Resonance Imaging Receive-Only Coil for HBO2 Studies

**DOI:** 10.3390/s22166076

**Published:** 2022-08-14

**Authors:** Azma Mareyam, Erik Shank, Lawrence L. Wald, Michael K. Qin, Giorgio Bonmassar

**Affiliations:** 1A. A. Martinos Center for Biomedical Imaging, Massachusetts General Hospital, Harvard Medical School, Charlestown, MA 02129, USA; 2Department of Anesthesia, Massachusetts General Hospital, Harvard Medical School, Boston, MA 02115, USA; 3Office of Naval Research, Arlington, VA 22203, USA

**Keywords:** brain imaging, hyperbaric research, fire, safety, RF, MRI

## Abstract

The paper describes a new magnetic resonance imaging (MRI) phased-array receive-only (Rx) coil for studying decompression sickness and disorders of hyperbaricity, including nitrogen narcosis. Functional magnetic resonance imaging (fMRI) is noninvasive, is considered safe, and may allow studying the brain under hyperbaric conditions. All of the risks associated with simultaneous MRI and HBO2 therapy are described in detail, along with all of the mitigation strategies and regulatory testing. One of the most significant risks for this type of study is a fire in the hyperbaric chamber caused by the sparking of the MRI coils as a result of high-voltage RF arcs. RF pulses at 128 MHz elicit signals from human tissues, and RF sparking occurs commonly and is considered safe in normobaric conditions. We describe how we built a coil for HBO2-MRI studies by modifying an eight-channel phased-array MRI coil with all of the mitigation strategies discussed. The coil was fabricated and tested with a unique testing platform that simulated the worst-case RF field of a three-Tesla MRI in a Hyperlite hyperbaric chamber at 3 atm pressure. The coil was also tested in normobaric conditions for image quality in a 3 T scanner in volunteers and SNR measurement in phantoms. Further studies are necessary to characterize the coil safety in HBO2/MRI.

## 1. Introduction

Hyperbaric oxygen therapy (HBO2) [1] is a Food and Drug Administration (FDA)-approved treatment for certain neurological conditions (e.g., decompression sickness [2], carbon monoxide poisoning [3], intracranial abscess [4], and), and is postulated to benefit in others (stroke [5], cerebral palsy [6], traumatic brain injury [7], and so on). Furthermore, neuroimaging can potentially shed light on neurological conditions that could disable or even injure SCUBA or navy divers (e.g., nitrogen narcosis [8], oxygen toxicity [9], and so on). Magnetic resonance imaging (MRI) and functional magnetic resonance imaging (fMRI) [10] have rapidly gained acceptance as “the neuroimaging gold standard” for diagnosing and evaluating neurologic conditions. MRI’s non-invasiveness and avoidance of ionizing radiation may allow elucidation of the human nervous system’s fine anatomic and functional nuances during HBO2 (Figure 1). Hyperbaric chambers provide oxygen administration in a manner that has few side effects [11]. Combining these two technologies (HBO2 and MRI) could potentially reveal other conditions that HBO2 and fine-tuning established HBO2 therapies can treat.

However, MRI has not been utilized in the study of HBO2 because of the very real hazards raised by these two challenging environments [12]. The hyperoxic high-pressure environment of HBO chambers and the powerful magnetic fields of the MRI scanner are traditionally incompatible. There are very real risks of mechanical injury to the integrity of most chambers by magnetic forces, space, and access challenges to the patients in both monoplace chambers and MRI scanners, as well as—likely the most concerning—the risk of fire from RF MRI-generated arcing (see Appendix A) in a hyperoxic environment. Despite the improvements in hardware designs of MRIs, a type of noise called “spike noise” is a relatively common, albeit infrequent, artifact in many, if not all (at one point or another), MRI systems [13,14]. There are many different causes behind spiking, but arcing is the underlying mechanism. Spiking can occur from static electricity build-up in a dielectric (e.g., clothing) or friction as a result of vibration (e.g., metal on metal) [15] and movement of gradient electrical cables [16], and issues with the presence of foreign objects and debris can result in arcing in the RF coils or coil plugs [17]. As static electricity is controlled in the chamber by high humidity and the gradient cables are located outside the chamber, we focused this paper on designing an arcing-free phased-array magnetic resonance imaging receive-only (Rx) coil for safety.

This paper present a new MRI phase array coil as a critical technological approach to mitigate the risks and enable MRI and fMRI studies to be performed under hyperbaric conditions. We have developed this new phased-array coil to allow for an increased understanding of HBO2 mechanisms and it serves as a clinical tool to evaluate the efficacy of HBO2 (Figure 1). The Appendix A detail the risks of operating a hyperbaric chamber in the MRI environment.

## 2. Materials and Methods

### 2.1. Eight-Channel Phased-Array Head Coil and the Preamplifier Interface Board

The coil was designed (Figure 2 top) to fit into an MRI-compatible hyperbaric chamber, with the subject’s head lying comfortably inside the coil helmet. The eight-channel coil helmet (L × W × H: 26 × 21 × 26 cm) was a single piece. The helmet was sized from a 3D surface reconstruction of average segmented MPRAGE and fabricated with a 3D printer in polycarbonate [18]. The eight-channel receive array had eight elliptically shaped overlapped loops to reduce the coupling between the nearest neighbor elements. The pre-amp decoupling strategy provides additional decoupling between the nearest neighbor elements [19]. Each element consisted of 16 AWG wires with six or seven evenly spaced capacitors. All individual elements were tuned to 123.25 MHz (i.e., the Larmor frequency of a 3 Tesla Siemens MRI system) and matched to a loaded impedance of 50 Ω to minimize the noise figure of the pre-amplifiers.

The impedance matching was performed using a lattice balun at the drive point to each coil element (Figure 2 top). The lattice balun also helps in reducing common mode currents on the cables, which is especially important for the 47 cm cable length between the coil and the interface box. For protection during transmission of the B1 field from the transmit body coil of the MRI system, the active detuning circuit was formed across the matching capacitor using an inductor and PIN diode to create a node of high impedance in the coil loop (Figure 2 top). During transmission (i.e., transmit mode), a DC current is fed in each element, resulting in a forward biasing of the PIN diode (protecting the pre-amplifiers from the high power), and the resonant parallel LC circuit inserts a high impedance in series with the coil loop, blocking the B1-induced current from flowing in each element. For additional protection during transmission, we added fuses in each of the eight coil elements and a floating cable trap on the eight bunched cables, which provides further common mode suppression [20]. The cable trap requirement is related to avoiding having cables with a length of λ/8 or greater that would otherwise act as an antenna and introduce RF noise and dangerous power levels.

An eight-channel pre-amplifier interface board (Figure 3) was built as an interface between the helmet coil and the 3T Skyra scanner. As this board needs power and contains elements that may heat up, we decided for safety reasons to place it outside the HBO2 Hyperlite chamber in the proximity of the window under normobaric conditions. The interface box had eight Siemens Trio 3T pre-amplifiers noise matched to 50 Ω. Additional cable traps were placed before the pre-amps for protection during transmission. A DC current of 100 mA to drive the eight PIN diodes at the input of each coil element was provided by the interface board. RF chokes of 1 μH were used to block RF currents induced in the DC line. A power supply (10 V) was also provided to each of the pre-amplifiers. Blocking capacitors (1000 pF) were used at the input of each pre-amplifier to prevent DC current from flowing and preventing saturation.

The coil was coated with parylene because it has the following: (1) Superior barrier properties against water vapor and moisture, which are quite common in HBO2, to isolate the coil from the subject and improve the coil SNR. (2) Parylene has good dielectric properties. It has a low dielectric constant and dissipation factor. A low dielectric constant means that, when exposed to a high electric field, it is harder to break down the material when sparking than it would be with a high dielectric constant. It also has high dielectric strength, allowing our RF signals to pass through without being lost or absorbed in the material as it will remain “transparent” to RF (up to 55 GHz). A 5 μm Parylene thickness was found to have a breakdown voltage or dielectric strength of 1.1 kV (nominally 6.8 kV/mil) up to frequencies of approximately 100 GHz [21,22]. (3) A biostable, biocompatible coating to avoid subject rashes; FDA-approved for implant applications. (4) Highly corrosion resistant; thus, we can clean the coil and disinfect it using standard cleaning solutions (especially for the application of harsh sterilization chemicals that will be used with the animal coil after each study). (5) Stable to oxidation up to 350 °C, preferable in highly oxygen-rich environments. (6) Low coefficient of friction (will help avoid static electricity sparks). (7) Very low permeability to gases so it will protect the underlying coil circuitry. (8) Transparency (may help us see through to help with future coil repairs). (9) Parylene C can withstand long-term high temperatures (i.e., 80 °C) as it is not a thermoplastic, necessary characteristic for safety. (10) Parylene vacuum coating encapsulates all components and produces a barrier against fuse/capacitors that may potentially burn out during scanning. Parylene is applied via vapor phase vacuum deposition and progresses evenly, covering all exposed surfaces with a thin, uniform, and pin-hole-free film. Because it is gas, it can reach into the smallest nooks and crannies that other liquid coatings cannot reach.

Finally, all of the conductive materials (e.g., copper wires and capacitors) were coated with a flame retardant two-part epoxy with 500 V/mil high dielectric strength (9200 FR, MG Chemicals, Burlington, Canada). The coating was visually and electrically inspected. The electrical inspection was performed with a Hipot Tester (V74, Virtek Corporation, Poway, CA, USA) by connecting one terminal to each of the eight coil loops (by running multiple tests for each loop) and the other to a copper tape on top and running the full length of the epoxy coating. The Hipot tester was programmed with a voltage level of 5 kV (DC), the DUT was capacitive and isolated, the ramp time was 30 s, and the dwell was 10 s, while the current limit was set to I_MAX_ = 20 µA, which are very conservative and standard settings.

### 2.2. RF Safety Testing

The coil passed the standard AA Martinos Center coil test, which consists of the following:Cover protection. The cover protection test checks if the cover protects the subject from touching any electrical parts of the coil.Fuses and/or passive circuits The fuses test checks if fuses were present in the coil.Availability of the SAR parameters. The SAR test parameters were included in the coil files located on the scanner.Detuning during the transmitting phase. The detuning during the transmitting phase checks whether the active detuning of the receive coil during transmitting is sufficient to prevent the high-voltage RF power from leaking into the receive coil.Heating and performance test. The heating and performance test checks the heating on the surface of the RF receive coil due to RF or gradient coil switching by running the default Siemens pulse sequences, namely grad-free pulse or fidseq. The grad-free pulse sequence parameters were selected with the number of averages = 100, TR = 2.5 s, and five measurements running for 5 min. The fidseq sequence parameters were selected to create an applied B1 field of the Body coil with 30 μT and a duty cycle of 5% over 15 min. The change in surface temperature ∆T before and after every measurement must be ≤4 °C.

In order to complete the RF-simulation test, a requirement for HBO2/MRI scanning clearance at the AA Martinos Center, the head coil was adapted to fit in the Hyperlite HBO2 (SOS Hyperlite Ltd., Isle of Man, UK) chamber. A double-ring structure was built to hold the braided Vectran soft shell on top of either of the two coils in place while the Hyperlite was being inflated. The two rings were made of acrylic plastic, rigid enough to protect the wires of the coils from the weight of the shell and flexible enough to be adjustable so as to be easily slid into the Hyperlite chamber while deflated. The Hyperlite chamber was retrofitted to become MRI-compatible by eliminating all of the ferromagnetic components and was adapted into the Hyperlite window through a set of 3D-printed or machined plastic adaptors (Figure 4A–C). As it did not fit in the Siemens Skyra body coil without major modifications, a separate Siemens Skyra body coil with the gradients set for shielding was used in this experiment and referred to as the “test rig”.

The Hyperlite was then inflated at 3 ATA with a mixture of medical air and Argon through the semirigid tubing (Figure 4D) connected to the newly developed console for the hyperbaric MRI compatible system (not shown). The partial pressure of approximately 200 Pa of Argon promotes glow when discharges may occur at irradiated RF power as low as 55 W [23]. Thus, we introduced Argon to increase the sensitivity to arcing detection. The Hyperlite with the HBO2 head coil was then slid into the Skyra body coil (i.e., the RF-simulator, see Figure 4D). The Skyra body coil was located inside a remote MRI bay, and testing was performed when no other person was around during a night-time slot. The Skyra body coil was connected to the RF amplifier of the 3 T Siemens Skyra by disconnecting the RF coax cable from the scanner and connecting the test rig instead. A special Siemens maintenance sequence was run for 6 h in several sessions (to allow for the RF amplifier to cool off), which allowed us only to use the RF amplifier while not turning ON any other components (e.g., gradient coils). With the light of the bay switched off, a highly sensitive thermal imaging camera (OSXL-101, Omega Engineering, Inc., Stamford, CT, USA) inside a special stainless steel protective enclosure (OSXL-100-PE) recorded any potentially unsafe infrared glow in the chamber, a method used in the design of arcing fault detectors [24]. The sequence was set to output the full power or 100% specific absorption rate (SAR) to test the worst-case scenario, as the 100% SAR is the maximum FDA-approved dosimetric power that can be safely delivered to humans.

### 2.3. MRI Imaging

#### 2.3.1. Phantom Imaging

The SNR map and noise covariance matrix had the following parameters: TR = 9.1 ms, TE = 4.8 ms, and averages = 4 [25].

#### 2.3.2. Human Imaging

The test MRI images were collected using the HBO2 coil (see above) in a Siemens Tim Trio at 3 Tesla using both a standard Siemens phantom and a healthy volunteer. The human study was conducted at the AA Martinos Center at MGH and was fully approved by the Partners Health System Institutional Review Board (IRB). BOLD images were acquired (TE = 56 ms, TR = 3 s, resolution = 3 mm (isotropic)) as well as T1 images (TE = 93 ms, TR = 9 s, TI = 2.5 s, resolution = 1.5 mm (isotropic)). The field map images were acquired with the parameters TE = 4.22 ms, TR = 0.5 s, and resolution = 1.5 mm (isotropic). T1-weighted images were acquired using an MPRAGE sequence with the parameters TE = 3.45 ms, TR = 2.53 s, TI = 1.1 s, and resolution = 1.5 mm (isotropic). Magnetic resonance angiography (MRA) had the following parameters: TE = 3.59 ms, TR = 20 ms, and resolution = 1.5 mm (isotropic).

## 3. Results

Each element shows a Q unloaded-to-loaded (with a human head) ratio of ~170/30. All elements were tuned to 123.25 MHz and matched to a loaded impedance of 50 Ω to minimize the noise figure of the pre-amplifiers. The bench measurements of the RF coil showed the s-parameters of all eight individual coil elements (see Appendix A). The S21 loaded between neighboring elements ranged from −18 dB to −12.5 dB. S11 reflections showed the elements tuned and matched to 50 Ω. These were typical RF receive coil parameters.

The following tests were performed in normobaric conditions to test the reliability and quality of the fabricated coils. The image quality tests of the RF coil (Figure 5) were the SNR map and the noise correlation matrix, which were performed on a saline solution cylindrical phantom. Figure 5A shows the sensitivity receive profile SNR map of the axial slice image at 3 T with the highest SNR around the surface of the phantom. The noise correlation matrix (Figure 5B) reveals low coupling or mutual inductance between the receive coil elements owing to the correct coil overlap design and the proper pre-amp decoupling. Finally, the coating does not seem to reduce the SNR (Appendix A).

The RF safety tests showed that the coil was intact, with all fuses working, and passed all eight Hipot tests. The detuning during the transmitting phase measurement recorded in the Siemens logfile (e.g., RFSWDHistoryListNew) transmits powers with and without the coil on the patient bed. These powers had a ratio of 1.009, indicating that the RF power leaving the RF power amplifier and delivered to the patient is within the safety limits. Furthermore, the gradient heating test after running the grad-free pulse sequence showed that the change in surface temperature ΔT before and after the test was 0.6 °C (16.1 °C vs. 16.7 °C), which was well below the 4 °C limit of heating. The heating during transmitting, after setting up the transmit voltage to 1.5 times the reference voltage (Vref = 335.5 V) and running the measurement for a duty cycle of 5% and for 15 min, yielded the change in surface temperature ΔT before and after the test of 2.2 °C (16.7 °C vs. 18.9 °C), which was well below the 4 °C limit of heating.

The safety at 3 Tesla of the coil was tested by retrofitting it into a Hyperlite HBO2 chamber (Figure 4A–C) and, after pressurization at 3 ATA, it was inserted into a test rig (Figure 4D). The RF safety test did not reveal the presence of arcing over the entire course of the six-hour run.

The following image acquisitions were performed on a human volunteer. Figure 6A shows the amplitude and, in Figure 6B, the phase of a field map is shown. Figure 6B,C illustrates the HBOT coil’s ability to acquire high-quality T1-weighted images. The T1-weighted images also allowed for cortical surface reconstruction (Figure 7 top) using a tool called “FreeSurfer” [26], which enabled us to obtain an automated, accurate, and explicit representation of the cortical surface in our volunteer, requiring no manual intervention. These procedures allow the routine cortical surface-based analysis of areas in the inflated model (Figure 7 middle) and visualization methods in functional brain imaging (Figure 7 bottom). The latter was obtained by a subject staring at a flickering checkerboard while being scanned at 3 T fMRI. MR angiography (MRA) images (Figure 8, top) show the ability of the HBOT coil to evaluate blood vessels and help identify abnormalities. Finally, diffusion tensor images (DTIs) (Figure 8, bottom) using anisotropic diffusion enabled us to estimate the axonal (white matter) organization of the brain.

## 4. Discussion

The work of standard societies such as the ASTM has led the FDA to develop recommendations and significantly evaluate the safety of devices in the MRI environment. However, a specific safety body of standards that can be used to determine the safety of HBO2 in the MRI environment is lacking; in this paper, we have described how to bridge the FDA’s regulatory recommendations to the case of HBO2 chambers by suggesting specific mitigation strategies. Furthermore, we fabricated a new coil for HBO2-MRI studies by modifying the design of a standard eight-channel phased-array MRI coil. The coil was fabricated and tested with a unique testing platform that simulated the worst-case RF field of a 3 Tesla MRI in a Hyperlite hyperbaric chamber at 3 atm pressure. Further testing is required to demonstrate the complete safety of the coil for 3 T/3 atm imaging. The coil was also tested in normobaric conditions for image quality in a 3 T scanner in volunteers and SNR measurement in phantoms. The fMRI images showed the potential ability to measure the brain’s BOLD activity, which may help uncover the brain areas involved in nitrogen narcosis or oxygen toxicity, which may reveal the mechanisms behind various gas toxicities at depth. The coils allow for a very detailed and high-quality MRA, which would enable studies of how decompression sickness affects blood vessels, given that MRA is used in acute ischemic stroke detection. Finally, the new HBO2 coil allowed acquiring DTIs to detect abnormalities in brain tissue caused by multiple sclerosis, stroke, Alzheimer’s disease, and brain tumors. DTI could monitor the HBO2 therapeutic effects on white matter diseases caused by decompression sickness. As DTI can characterize microstructural changes with neuropathology, it may also monitor the concentration of microbubbles in the central nervous system during HBO2 treatment.

## 5. Conclusions

For many years, hyperbaric oxygen therapy (HBO2) has provided safe and effective FDA-approved treatments for several neurological diseases (e.g., decompression sickness, carbon monoxide poisoning, and intracranial abscess) and offers a valuable environment for studying conditions that may affect SCUBA diving (e.g., nitrogen narcosis and oxygen toxicity). In this paper, we have designed and tested the first HBO2 phased-array coil for safety and efficacy for 3T MRI. The main differences to standard MRI coil design consisted of the new conformal coating that provided high dielectric strength and current limiting components. Both changes were introduced to avoid arcing/fire hazards in the chamber. Mechanical modifications were applied to the phased-array helmet coil design to fit the Hyperlite’s window (Figure 4C). Furthermore, the positioning of the pre-amp outside the HBO2 chamber was performed again to reduce potential fire hazards. The proposed phased-array coil for HBO2/MRI may enable neuroimaging, which in turn could help elucidate the mechanism of action of HBO2 for various diseases or conditions.

## Figures and Tables

**Figure 1 sensors-22-06076-f001:**
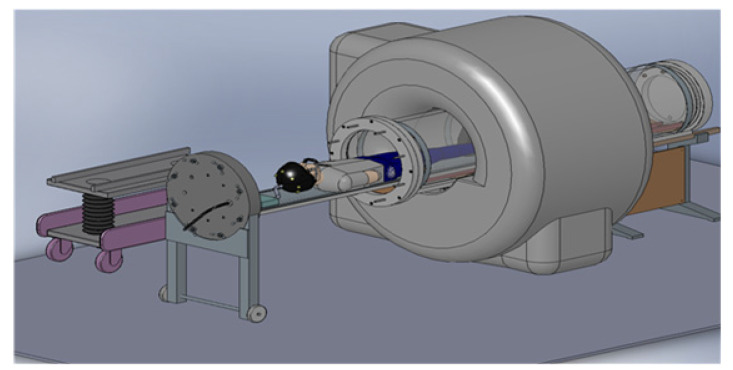
The proposed hyperbaric MRI compatible chamber and the general layout.

**Figure 2 sensors-22-06076-f002:**
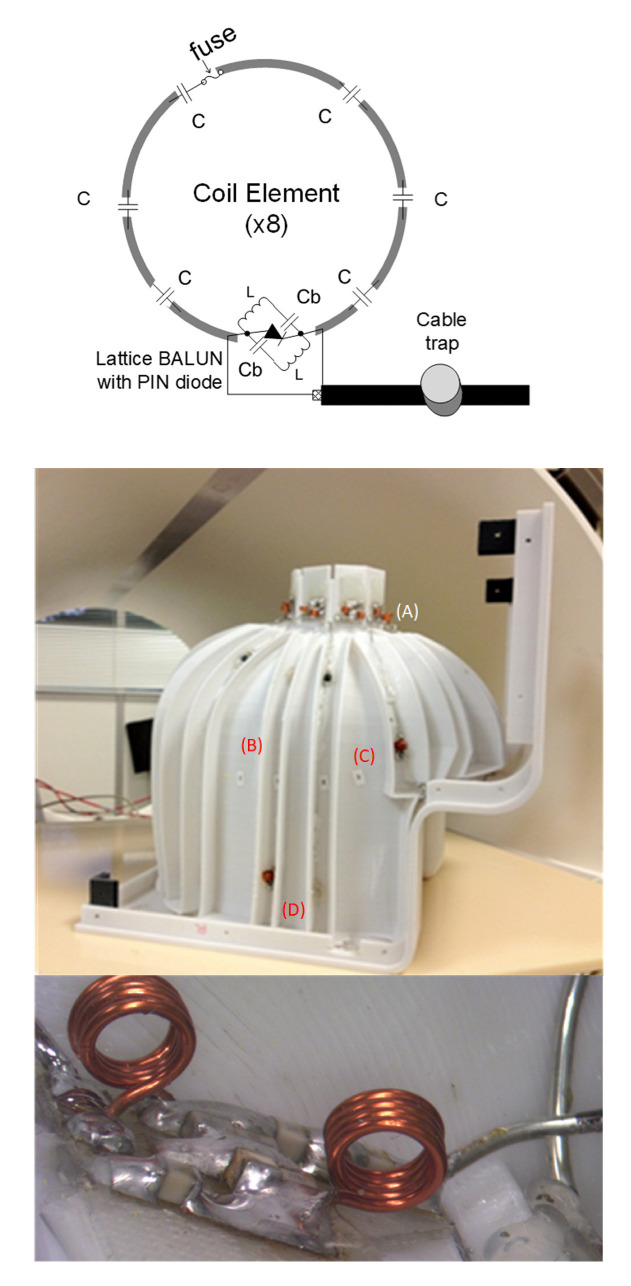
(**Top**) Schematics of the RF coil and the active decoupling. (**Middle**) The 3D-printed eight-channel phased-array helmet: (A) lattice balun with a PIN diode for detuning during transmission, (B) copper wires loops, (C) distributed capacitors, and (D) fuses. (**Bottom**) A detailed image of the lattice balun and every soldering was inspected and photographed.

**Figure 3 sensors-22-06076-f003:**
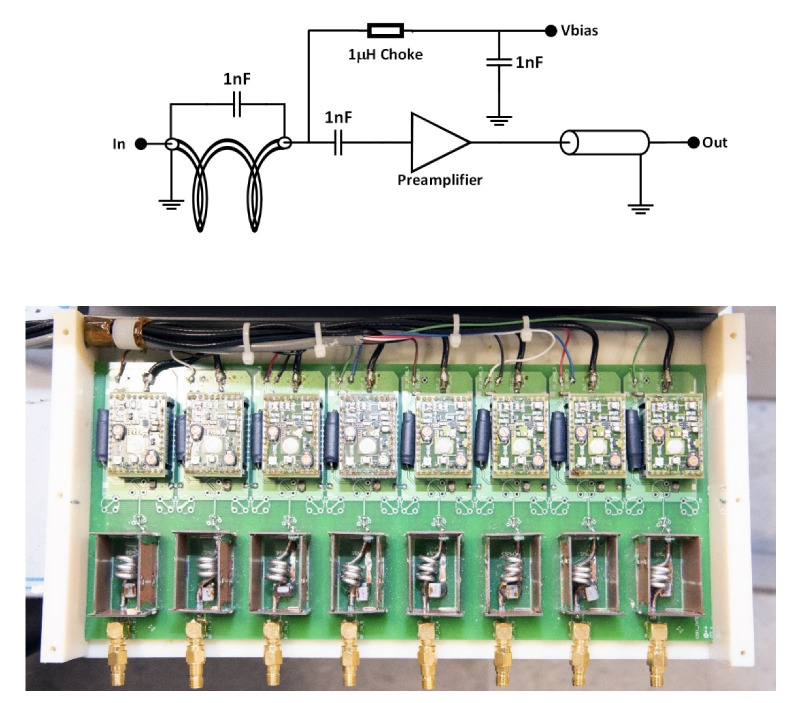
(**Top**) Schematics of the eight-channel pre-amplifier interface board. (**Bottom**) Images of the pre-amp board populated with components and connections.

**Figure 4 sensors-22-06076-f004:**
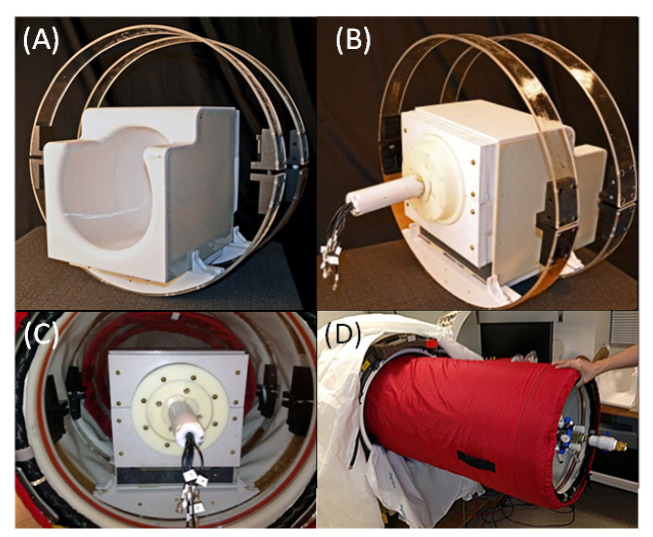
Images of the HBO2-compatible MRI coil with special mechanical modification with respect to the standard commercial MRI coil to allow it to fit into the Hyperlite window (**A**,**B**), during pressurization testing (**C**), and while being inserted in a Siemens body RF coil (**D**).

**Figure 5 sensors-22-06076-f005:**
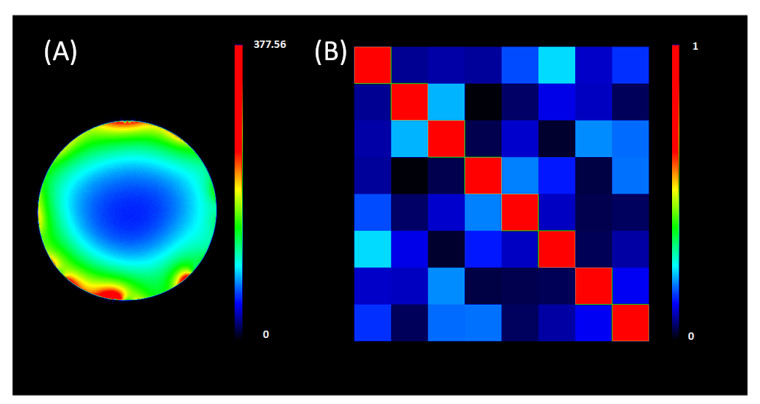
Normobaric 3 Tesla MR image quality tests [25] of a saline phantom: (**A**) signal-to-noise ratio and (**B**) noise covariance matrix.

**Figure 6 sensors-22-06076-f006:**
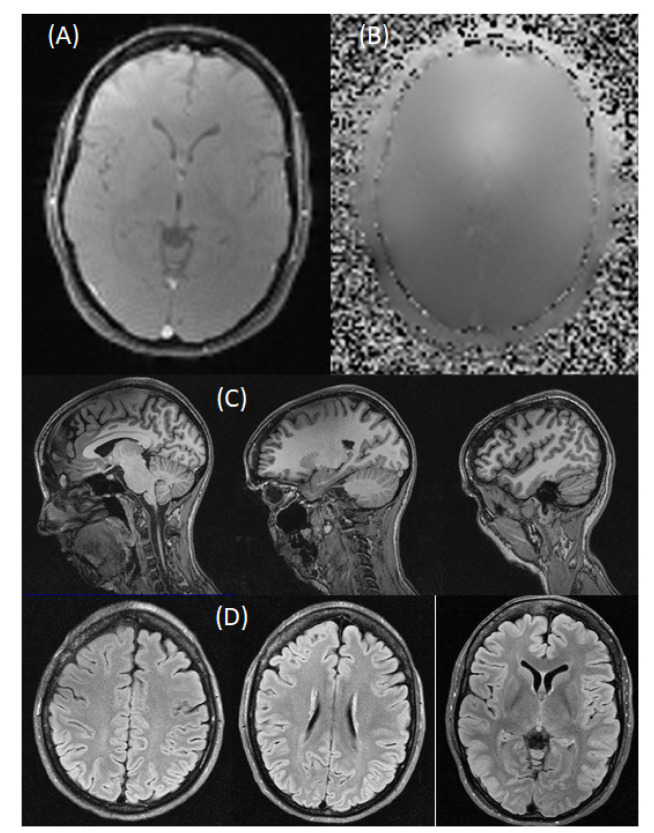
Three Tesla normobaric MR images. (Top) Field map images of magnitude (**A**) and phase (**B**), sagittal (**C**), and axial (**D**) T1-weighted MPRAGE images.

**Figure 7 sensors-22-06076-f007:**
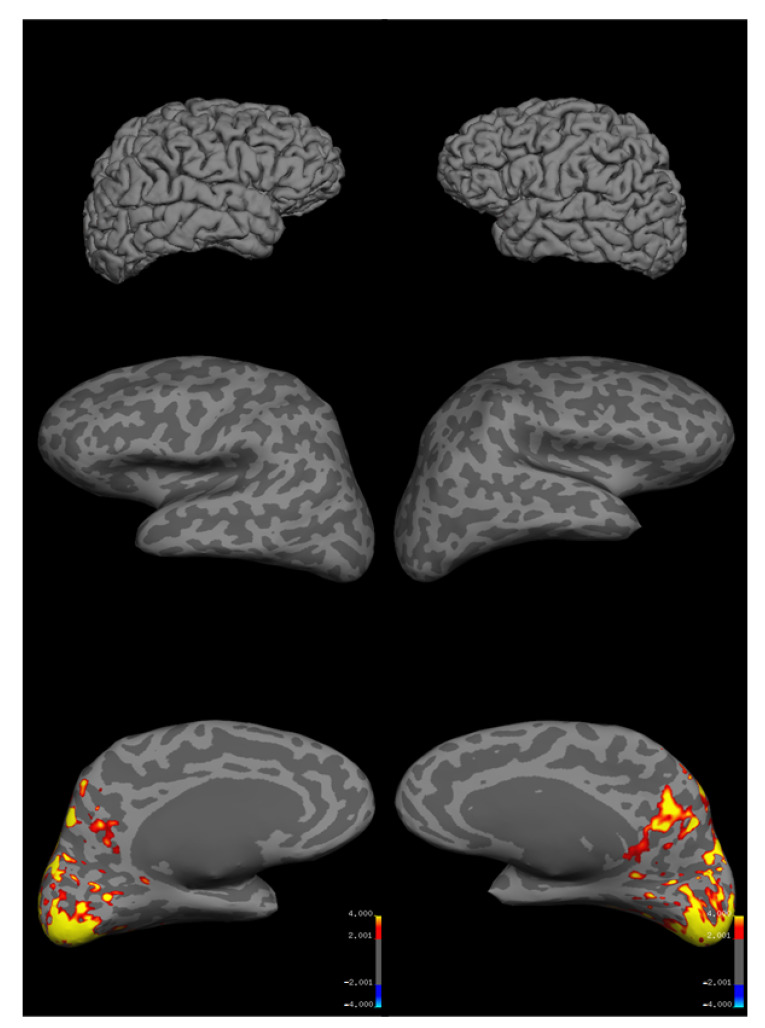
Three Tesla fMRI. Segmented brain from the T1 MPRAGE (**top**), inflated brain (**middle**), and functional BOLD images projected onto the inflated brain surface (**bottom**).

**Figure 8 sensors-22-06076-f008:**
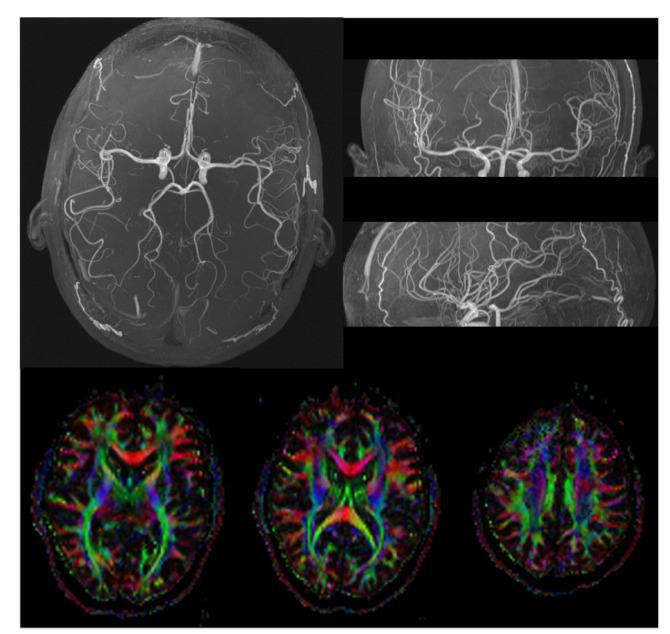
(**Top**) Images of magnetic resonance angiography (MRA). (**Bottom**) Diffusion tensor imaging (DTI) with colors indicating directions as follows: red, left–right; green, anteroposterior; blue, superior–inferior.

## Data Availability

Data are available upon request because of HIPAA restrictions. The data presented in this study are available upon request from the corresponding author.

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
