# Peer review of "A New Phased-Array Magnetic Resonance Imaging Receive-Only Coil for HBO2 Studies"

_sensors, 2022, doi:10.3390/s22166076_

Round 1

Reviewer 1 Report

The manuscript “A NEW PHASED ARRAY MAGNETIC RESONANCE IMAGING COIL FOR HBO2 STUDIES” presents the development and testing of a parallel coil for MRI imaging in hyperbaric conditions.

The manuscript is not at a stage for a journal submission, and I suggest the senior authors to revise the text carefully before resubmitting. In this way several no sense expressions could be corrected.

I also suggest the authors to focus on the title topic and concentrate their efforts in describing design strategies, realization, and tests of the RF coil only, without the long confuse introduction mainly focused on generic discussion of risks and tests of the hyperbaric chamber in MRI environment.

Please increase the quality of some figures (electrical schemes in Fig. 2 and 3, in the latter symbols are not readable) and put care on their content: Fig. 5 presents a “sensitivity profile receive profile SNR map” (quoting the manuscript), where the colormap of the noise correlation is marked with a length scale.

The reader of this manuscript will also be interested in knowing more coil details (decoupling of non-overlapping loops, electrical scheme of active decoupling), if arching mitigation measures like special coating are really necessary, a performance comparison before/after coil coating as well as a comparison with a standard head parallel coil to know if reduced performances are a price to pay for working in hyperbaric environment. The above piece on information is much more valuable than reading that in T1-w images fat is bright, CSF is dark, Free-Surfer allows cortical surface reconstruction, what MRA and DTI are…

Author Response

The manuscript "A NEW PHASED ARRAY MAGNETIC RESONANCE IMAGING COIL FOR HBO2 STUDIES" presents the development and testing of a parallel coil for MRI imaging in hyperbaric conditions.

The manuscript is not at a stage for a journal submission, and I suggest the senior authors to revise the text carefully before resubmitting. In this way several no sense expressions could be corrected.

A.: We thank the Reviewer, and the manuscript has now been thoroughly reviewed and corrected, also based on the comments of the other reviewers.

I also suggest the authors to focus on the title topic and concentrate their efforts in describing design strategies, realization, and tests of the RF coil only, without the long confuse introduction mainly focused on generic discussion of risks and tests of the hyperbaric chamber in MRI environment.

 A.: The two figures mentioned (Figures 2 and 3) have been largely redone and now contain readable symbols and corrected colormap legends.

The reader of this manuscript will also be interested in knowing more coil details (decoupling of non-overlapping loops, electrical scheme of active decoupling), if arching mitigation measures like special coating are really necessary, a performance comparison before/after coil coating as well as a comparison with a standard head parallel coil to know if reduced performances are a price to pay for working in hyperbaric environment. The above piece on information is much more valuable than reading that in T1-w images fat is bright, CSF is dark, Free-Surfer allows cortical surface reconstruction, what MRA and DTI are…

A: In response to the Reviewer, we added a description of the active decoupling along with a new schematic in Figure 2. The text now reads: "The 8-channel receive array had eight elliptically shaped overlapped loops to reduce the coupling between the nearest neighbor elements. The pre-amp decoupling strategy provides additional decoupling between next nearing neighbor elements [14]. Each element consisted of 16 AWG wires with six or seven evenly spaced capacitors. All individual elements were tuned to 123.25MHz (i.e., the Larmor frequency of a 3 Tesla Siemens MRI system) and matched to a loaded impedance of 50 W to minimize the noise figure of the pre-amplifiers. The impedance matching was done by using a lattice balun at the drive point to each coil element (Figure 2 top). The lattice balun also helps in reducing common mode currents on the cables, which is especially important for the 47cm cable length between the coil and the interface box."

Finally, we added a comparison of before and after coating of a single loop coil (OD=9cm) to understand if there is any reduced performance. The results in Figure S6 show that the coating does not reduce the SNR.

Reviewer 2 Report

Hyperbaric oxygen therapy has shown efficacies in the treatment of a number of diseases as it aims to improve the supply of oxygen to the brain to activate the generation of new blood vessels and neurons. The goal of this study was to develop an eight channel phased array head coil integrated in a 3D printed Helmet.

Some suggestion to the authors:
Figure 3 is not clear, on top legend is not visible
at line 163
Under hyperbaric oxygen, images exhibited artifacts and temporal instability attributable also to fluctuating oxygen concentration from air and oxygen mixing near the imaging region. Have you consider this issue?
at line 342 Figure 6A shows the ampliture and Figure 6B the phase
In Figure 7 ad A,B,C (cited in the text) and not Top, middle, bottom

Author Response

Hyperbaric oxygen therapy has shown efficacies in the treatment of a number of diseases as it aims to improve the supply of oxygen to the brain to activate the generation of new blood vessels and neurons. The goal of this study was to develop an eight channel phased array head coil integrated in a 3D printed Helmet.

Some suggestion to the authors:
Figure 3 is not clear, on top legend is not visible
A: We thank the Reviewer for pointing this out, and Figure 3 has now been updated to allow for better legibility.

at line 163
Under hyperbaric oxygen, images exhibited artifacts and temporal instability attributable also to fluctuating oxygen concentration from air and oxygen mixing near the imaging region. Have you consider this issue?

A: We thank the Reviewer for this note. We did make experiments with a smaller chamber (B11, see attached PDF file) with 100% oxygen in the chamber and a single channel surface coil but only for safety testing, and we did not notice any changes in adjustments or image quality, this is because the electrical and magnetic properties are very similar (i.e., O2 vs. air). For instance, the relative dielectric constant and the magnetic permeability of air are: εr= 1.0004947 and μv=3.6·10-7 H/m while for oxygen: εr= 1.0005364 and μv=3.73·10-7 H/m. We added the following sentence to the mentioned paragraph, which was moved to the Supplementary materials following Reviewer 1 request: "However, the various changes in gas concentration are not likely cause of artifacts as the electrical and magnetic properties of air and oxygen are very similar (...)" (line 155, Supplementary Material).

at line 342 Figure 6A shows the ampliture and Figure 6B the phase

A: We thank the Reviewer for picking this up, the sentence has now changed to: "Figure 6A shows the amplitude and in Figure 6B the phase of a field map."

In Figure 7 ad A,B,C (cited in the text) and not Top, middle, bottom

A: We thank the Reviewer for finding this mislabeling. The sentence now reads: "T1-weighted images also allow for cortical surface reconstruction (Figure 7 top) using a tool called "FreeSurfer" 1, which enabled us to obtain an automated and accurate, and explicit representation of the cortical surface in our volunteer, requiring no manual intervention. These procedures allow the routine cortical surface-based analysis of areas in the inflated model (Figure 7 middle) and visualization methods in functional brain imaging (Figure 7 bottom). The latter was obtained by a subject staring at a flickering checkerboard while being scanned at 3T fMRI."

Reviewer 3 Report

In this paper, the authors constructed a new MRI probe system which incorporated a commercial hyperbaric chamber. The authors paid special cares to avoid rf arcing in the O2 containing hyperbaric chamber. I have the following comments:

1. The subject of this paper is the probe arcing (not "arching" as often appears in the text). But the author did not tackle the problem directly. The physics of arcing have been studied for a long period by plasma physicists but not by nmr or mri researchers. Please look at "electrical breakdown", "electric discharge in gases" in Wikipedia. Although the dc electric discharge is explained there, the physics is similar for the rf discharge. See the following paper on the experimental and simulation works.

H. B. Smith, C. Charles, and R. W. Boswell, Physics of Plasma, 10, 875-881 (2003).

Above the breakdown voltage the free electrons in the sample obtain enough energy from rf electric fields to cause ionizations of nearby gas molecules and to produce more free electrons and produce more free electrons (an electron avalanche). So it may be important where in the MRI coil, the electric field becomes so large and an arcing takes place. Now it may not be difficult to calculate the electric field in MRI coils and it may be possible to predict arcing in the coils. There may be a coil design which may show less arcing.

2. The authors did not cite any papers on the receiver/transmitter and preamplifier systems. What are the differences from the existing designs?

3. The careful spell checks are required including "arching", "hyperbolic chamber"(line 187). The paragraph starting from line 279 may contain some errors.

4. The parylene coating may work for the dc breakdown. However is it also effective to the rf breakdown?  

Author Response

In this paper, the authors constructed a new MRI probe system which incorporated a commercial hyperbaric chamber. The authors paid special cares to avoid rf arcing in the O2 containing hyperbaric chamber. I have the following comments:

1. The subject of this paper is the probe arcing (not "arching" as often appears in the text). But the author did not tackle the problem directly. The physics of arcing have been studied for a long period by plasma physicists but not by nmr or mri researchers. Please look at "electrical breakdown", "electric discharge in gases" in Wikipedia. Although the dc electric discharge is explained there, the physics is similar for the rf discharge. See the following paper on the experimental and simulation works.

H. B. Smith, C. Charles, and R. W. Boswell, Physics of Plasma, 10, 875-881 (2003).

Above the breakdown voltage the free electrons in the sample obtain enough energy from rf electric fields to cause ionizations of nearby gas molecules and to produce more free electrons and produce more free electrons (an electron avalanche). So it may be important where in the MRI coil, the electric field becomes so large and an arcing takes place. Now it may not be difficult to calculate the electric field in MRI coils and it may be possible to predict arcing in the coils. There may be a coil design which may show less arcing.

A: We thank the Reviewer for pointing out the typos and for the excellent physical explanation of the arcing phenomenon. We have corrected the spelling, we introduced the proposed reference ad the following text in the Supplementary Material, in the paragraph where the arcing was initially mentioned and now moved to address Reviewer 1 concerns: "Arcing is a change of neutral gas from an insulation state to a conducting state, which is an insulation breakdown that will occur at a certain electrical field E. Generally, the voltage at which the arcing occurs is defined as the dielectric strength of the gas. Breakdown in gases is generally produced by impact ionization of the gas molecules at DC, while it is produced by electron diffusion at RF 13."

2. The authors did not cite any papers on the receiver/transmitter and preamplifier systems. What are the differences from the existing designs?

A:  In order to address this point, we added a sentence citing one of our previous coil designs: ". The helmet was sized from a 3D surface reconstruction of average segmented MPRAGE and fabricated with a 3D printer [13]." Furthermore, now the conclusions summarize the key differences with traditional coil designs: "The main differences to standard MRI coil design consisted of the new conformal coating that provided high dielectric strength and current limiting components. Both changes were introduced to avoid arcing/fire hazards in the chamber. Mechanical modifications were applied to the phased array helmet coil design to fit the Hyperlite's window. Furthermore, the positioning of the pre-amp outside the HBO2 chamber was performed again to reduce potential fire hazards."

3. The careful spell checks are required including "arching", "hyperbolic chamber"(line 187). The paragraph starting from line 279 may contain some errors.

A: We thank the Reviewer for catching the typo, which has been corrected. The paragraph in the old line 279 has also been rephrased as follows; "The Hyperlite was then inflated with a mixture of air and Argon through the semi-rigid tubing (Figure 4) connected to the newly developed console for the hyperbaric MRI compatible system (not shown). The partial pressure of approximately 200 Pa of Argon promoted glow when discharges occur at low 55W RF power{Winchester, 2004 #86668}. The Skyra body coil was placed inside a remote MRI Bay, and testing was performed when no other person was around during a nighttime slot. The test rig was connected to the RF amplifier of the 3 T Siemens Skyra by disconnecting the RF coax cable from the scanner and connecting the test rig instead. A special Siemens maintenance sequence was run for 6 hours in several sessions (to allow for the RF amplifier to cool off), which allowed us only to use the RF amplifier while not turning ON any other components (e.g., gradient coils, etc.). With the light of the Bay switched off, a highly sensitive thermal imaging camera (OSXL-101, Omega Engineering, Inc., Stamford, CT) inside a special stainless steel protective enclosure (OSXL-100-PE) recorded any potentially dangerous infrared glow in the chamber, a method used in the design arcing fault detectors [13]. The sequence was set to output the full power or 100% SAR to test the worst-case scenario."

4. The parylene coating may work for the dc breakdown. However is it also effective to the rf breakdown? 

A: The text has now been modified as follows:" 

  1. Parylene has good dielectric properties. It has a low dielectric constant and dissipation factor. A low dielectric constant means that when exposed to a high electric field, it is harder to break down the material when sparking than it would be with a high dielectric constant. It also has high dielectric strength, allowing our RF signals to pass through without being lost or absorbed in the material as it will remain "transparent" to RF (up to 55GHz). A 5 mm Parylene thickness was found to have a breakdown voltage or dielectric strength of 1.1kV (nominally 6.8kV/mil) up to frequencies of approximately 100GHz [16, 17]."

Reviewer 4 Report

In the paper, construction and test of an eight-channel phased-array coil for performing high-field magnetic resonance imaging in a hyperbaric chamber for oxygen therapy, for instance for patients suffering from nitrogen narcosis. Special attention is put on safety with respect to sparking due to high voltage radio-frequency arcs, which pose the danger of fire ignition in the high-pressure oxygen. The coil has been successfully tested under both high and ambient pressure conditions in conjunction with the most common imaging modalities of medical MRI.

To the best of my knowledge, the work is novel and original. The title of the paper is appropriate. The abstract summarizes the approach and the main results. The paper is well structured. In the introduction, the work is put into proper context to previously published research, and all the issues relevant for safety (including hyperbaric oxygen, static field effects, pulsed gradient field related risks, radio-frequency field effects, electrostatic arching, and image artifacts) are listed and discussed. In the Materials and Methods section 2, the coil configuration is comprehensively explained and illustrated with a number of instructive photographs. The results of rf reflection and transmission measurements, of rf safety tests and of MR image quality tests are described in detail in section 3. In the discussion, the results of the tests so far performed are presented and the still missing additional safety tests are outlined. In the conclusion, the results are briefly summarized and an outlook to medical applications is given.

In revision, I suggest addressing the following very minor points:

1) Line 34-35: the two mdoalities MRI and fMRI can be listed with “and”: “Magnetic resonance imaging (MRI) and functional magnetic resonance 34 imaging (fMRI) [10] have rapidly gained acceptance … ”.

2) Line 52: The sentence appears incomplete to me. Maybe add: “… understanding of HBO2 mechanisms, and serve as a clinical tool …”.

3) Line 71: I think you mean: “We will describe the risks imposed by each MR imaging component … ”.

4) Equation (1) on line 138: for physicists, greek letter omega implies that angular frequency 2*pi*f (in units of 1/s) is meant, whereas f denotes technical frequency (in units of Hz). I suggest to adhere to that convention. As you give the gyromagnetic ratio gamma in MHz/T, the formula should contain technical frequency f.

Author Response

In the paper, construction and test of an eight-channel phased-array coil for performing high-field magnetic resonance imaging in a hyperbaric chamber for oxygen therapy, for instance for patients suffering from nitrogen narcosis. Special attention is put on safety with respect to sparking due to high voltage radio-frequency arcs, which pose the danger of fire ignition in the high-pressure oxygen. The coil has been successfully tested under both high and ambient pressure conditions in conjunction with the most common imaging modalities of medical MRI.

To the best of my knowledge, the work is novel and original. The title of the paper is appropriate. The abstract summarizes the approach and the main results. The paper is well structured. In the introduction, the work is put into proper context to previously published research, and all the issues relevant for safety (including hyperbaric oxygen, static field effects, pulsed gradient field related risks, radio-frequency field effects, electrostatic arching, and image artifacts) are listed and discussed. In the Materials and Methods section 2, the coil configuration is comprehensively explained and illustrated with a number of instructive photographs. The results of rf reflection and transmission measurements, of rf safety tests and of MR image quality tests are described in detail in section 3. In the discussion, the results of the tests so far performed are presented and the still missing additional safety tests are outlined. In the conclusion, the results are briefly summarized and an outlook to medical applications is given.

A: The authors thank the Reviewer for the careful reading of the manuscript, insightful comments, and constructive remarks.

In revision, I suggest addressing the following very minor points:

1) Line 34-35: the two mdoalities MRI and fMRI can be listed with "and": "Magnetic resonance imaging (MRI) and functional magnetic resonance 34 imaging (fMRI) [10] have rapidly gained acceptance …".

A: We thank the Reviewer and the text was modified as follows: "Magnetic resonance imaging (MRI), and functional magnetic resonance imaging (fMRI), [10] have rapidly gained acceptance as "the neuroimaging gold standard" for diagnosing and evaluating neurologic conditions."

2) Line 52: The sentence appears incomplete to me. Maybe add: "… understanding of HBO2 mechanisms, and serve as a clinical tool …".

A:  We thank the Reviewer and the sentence now reads: "We hope that this technological breakthrough will allow for an increased understanding of HBO2 mechanisms and serve as a clinical tool to evaluate the efficacy of HBO2 (Figure 1)."

3) Line 71: I think you mean: "We will describe the risks imposed by each MR imaging component …".

A: Thanks to the Reviewer's comments, we rephrased the sentence as follows: "We will describe the potential risks associated with each MR imaging component, as this may be considered the more complex technology from an FDA regulatory perspective.

4) Equation (1) on line 138: for physicists, greek letter omega implies that angular frequency 2*pi*f (in units of 1/s) is meant, whereas f denotes technical frequency (in units of Hz). I suggest to adhere to that convention. As you give the gyromagnetic ratio gamma in MHz/T, the formula should contain technical frequency f.

A: Following the Reviewer's suggestion, the text was now changed to:

The RF is used to excite the spins of protons in the subject's body according to Larmor's frequency f: ... (1)

Round 2

Reviewer 1 Report

Please consider the PDF file

Author Response

I appreciate the efforts of the authors to improve the manuscript quality and readability.

A.: We would like to thank Reviewer #1 for his/her particularly insightful comments and careful/patient review work that has helped to improve the manuscript. Please find in the attached file the detailed point-by-point response to all of your comments.
